# The Physical Activity and Nutritional INfluences in Ageing (PANINI) Toolkit: A Standardized Approach towards Physical Activity and Nutritional Assessment of Older Adults

**DOI:** 10.3390/healthcare10061017

**Published:** 2022-05-31

**Authors:** Keenan A. Ramsey, Carel G. M. Meskers, Marijke C. Trappenburg, Maria Giulia Bacalini, Massimo Delledonne, Paolo Garagnani, Carolyn Greig, Victor Kallen, Nico van Meeteren, Natal van Riel, Nadine Correia Santos, Sarianna Sipilä, Janice L. Thompson, Anna C. Whittaker, Andrea B. Maier

**Affiliations:** 1Department of Human Movement Sciences, @AgeAmsterdam, Vrije Universiteit Amsterdam, Amsterdam Movement Sciences, Van der Boechorststraat 7, 1081 BT Amsterdam, The Netherlands; keenanramsey13@gmail.com (K.A.R.); c.meskers@amsterdamumc.nl (C.G.M.M.); 2Department of Rehabilitation Medicine, Amsterdam University Medical Center, VU University Medical Center, De Boelelaan 1117, 1081 HV Amsterdam, The Netherlands; 3Department of Internal Medicine, Section of Gerontology and Geriatrics, Amsterdam University Medical Center, VU University Medical Center, De Boelelaan 1117, 1081 HV Amsterdam, The Netherlands; m.trappenburg@amsterdamumc.nl; 4Department of Internal Medicine, Amstelland Hospital, Laan van de Helende Meesters 8, 1186 AM Amstelveen, The Netherlands; 5IRCCS Istituto delle Scienze Neurologiche di Bologna, Padiglione G, Via Altura, 3, 40139 Bologna, Italy; mariagiulia.bacalini@ausl.bologna.it; 6Department of Biotechnology, University of Verona, Strada Le Grazie 15, 37134 Verona, Italy; massimo.delledonne@univr.it; 7Department of Experimental, Diagnostic and Specialty Medicine, University of Bologna, Via Zamboni 33, 40126 Bologna, Italy; paolo.garagnani2@unibo.it; 8Clinical Chemistry, Department of Laboratory Medicine, Karolinska Institutet, Karolinska University Hospital, Alfred Nobels allé 8 141 52 Huddinge, 10316 Stockholm, Sweden; 9Personal Genomics S.r.l., Via Roveggia, 43B, 37136 Verona, Italy; 10School of Sport, Exercise and Rehabilitation Sciences, University of Birmingham, Birmingham B15 2TT, UK; c.a.greig@bham.ac.uk (C.G.); j.thompson.1@bham.ac.uk (J.L.T.); a.c.whittaker@stir.ac.uk (A.C.W.); 11MRC-Versus Arthritis Centre for Musculoskeletal Ageing Research, University of Birmingham, Copeman House, St Mary’s Court, St Mary’s Gate, Chesterfield S41 7TD, UK; 12Department of Microbiology and System Biology, The Netherlands Organization for Applied Scientific Research, Utrechtseweg 48, 3704 HE Zeist, The Netherlands; victor.kallen@tno.nl; 13Top Sector Life Sciences & Health (Health~Holland), Wilhelmina van Pruisenweg 104, 2595 AN The Hague, The Netherlands; meeteren@health-holland.com; 14Department of Anesthesiology, Erasmus Medical Center, Dr. Molenwaterplein 40, 3015 GD Rotterdam, The Netherlands; 15Department of Biomedical Engineering, Eindhoven University of Technology, Postbus 513, 5600 MB Eindhoven, The Netherlands; n.a.w.v.riel@tue.nl; 16Life and Health Sciences Research Institute (ICVS), School of Medicine, University of Minho, 4710-057 Braga, Portugal; nsantos@med.uminho.pt; 17ICVS/3B’s, PT Government Associate Laboratory, AvePark, Zona Industrial da Gandra S. Claudio do Barco Caldas das Taipas, 4806-909 Guimarães, Portugal; 18Faculty of Sport and Health Sciences, University of Jyväskylä, Building Viveca (Viv), Rautpohjankatu 8, 40700 Jyväskylä, Finland; sarianna.sipila@jyu.fi; 19Sport, Health & Exercise Research & Education (SpHERE), Faculty of Health Sciences and Sport, University of Stirling, Stirling FK9 4LA, UK; 20Department of Medicine and Aged Care, @AgeMelbourne, The Royal Melbourne Hospital, The University of Melbourne, 300 Grattan Street, Parkville, VIC 3052, Australia; 21Healthy Longevity Translational Research Program, Yong Loo Lin School of Medicine, National University of Singapore, Singapore 119228, Singapore; 22Centre for Healthy Longevity, @AgeSingapore, National University Health System, 10 Medical Drive, Singapore 117597, Singapore

**Keywords:** geriatric assessment, aged, nutrition, physical activity, standardization

## Abstract

Assessing multiple domains of health in older adults requires multidimensional and large datasets. Consensus on definitions, measurement protocols and outcome measures is a prerequisite. The Physical Activity and Nutritional INfluences In Ageing (PANINI) Toolkit aims to provide a standardized toolkit of best-practice measures for assessing health domains of older adults with an emphasis on nutrition and physical activity. The toolkit was drafted by consensus of multidisciplinary and pan-European experts on ageing to standardize research initiatives in diverse populations within the PANINI consortium. Domains within the PANINI Toolkit include socio-demographics, general health, nutrition, physical activity and physical performance and psychological and cognitive health. Implementation across various countries, settings and ageing populations has proven the feasibility of its use in research. This multidimensional and standardized approach supports interoperability and re-use of data, which is needed to optimize the coordination of research efforts, increase generalizability of findings and ultimately address the challenges of ageing.

## 1. Introduction

Ageing has a strong potential to influence and be influenced by multiple domains of health and lifestyle in older adults [1,2]. Focusing on nutrition and physical activity as lifestyle factors is important in research and clinical practice as they are modifiable and are both determinants of health outcomes [3,4,5]. The high prevalence of health conditions that cross-cut multiple domains, such as sarcopenia and frailty, requires an understanding of the interactions between physical activity and nutrition for prevention and treatment strategies [6,7,8].

Heterogeneous, multidimensional and therefore large datasets requiring cross-national research and data from different cohorts are important to compare and study the interactions between different health domains [9]. However, a lack of shared definitions, methodology and outcome measures across different settings and ageing populations makes it difficult to synthesize data, results and conclusions [10,11,12]. Further, fragmented assessment of individual lifestyle factors and health outcomes often overlooks the interaction between separate domains (e.g., the impact of nutrition on cognition). A multidimensional and standardized approach is required for data synthesis to understand the complexity of overlapping domains in the health of older adults.

The PANINI project is a European Commission Horizon 2020 Marie Curie-Skłodowska Innovative Training Network that aimed to: (1) develop a toolkit of best-practice measures for assessing the health domains of nutrition and physical activity, plus related key concepts, in older adults with a multidimensional design that deliberately accounts for potentially overlapping and interacting domains; (2) implement this toolkit for data collection across all new PANINI research projects; and (3) use the toolkit to create a shared dataset comprising standardized measures that spans across the various aspects involved in the health of older adults measured in different ageing populations. This article describes the development of the PANINI Toolkit, including its content and application within the PANINI consortium.

## 2. Materials and Methods

### 2.1. Design

The PANINI Toolkit was developed in 2016 in two consecutive consensus meetings of leading experts across Europe in the field of ageing, physical activity, and nutrition (including health scientists, biologists, geneticists, epidemiologists, computer scientists, clinicians, nutrition and exercise scientists, and psychologists) who formed the supervisory board of the PANINI consortium. Key areas of healthy ageing were defined as domains to develop the toolkit. The PANINI Toolkit focuses on two domains, nutrition and physical activity/physical performance, but also encompasses domains of socio-demographics, general health, anthropometrics, and psychological and cognitive health. Domains were predefined by the consortium prior to tool and measure selection.

### 2.2. Tool and Measure Selection

Measures across all domains were collated based on current use, previous use (existing datasets) and expert opinion discussion of which measures should be used. Best-practice measures within these domains were ultimately chosen for inclusion in the toolkit based on validity, reliability and ease of use. Gold-standard measures (and measures that were correlated highly with their gold standard) were selected with preference for inclusion, as well as measures that were used broadly in large cohort studies or widely and internationally used in clinical practice. Consensus was reached through discussion, and any disagreements were resolved through evaluation of the aforementioned consideration factors, and similar tools/measures were included when the added value of including both could be justified and agreed upon. The customized PANINI Questionnaire was created to assess factors that were deemed important but not covered by other included tools. The customized PANINI Questionnaire, made up of 35 questions, is used to assess the domains of socio-demographics and general health, as well as the subdomains mobility and falls. Questions were designed based on consensus and with the specific intention of having broad applicability in diverse cohorts and to take into account population differences. All tools and measures were standardized into a protocol and a case report form.

## 3. Results

### 3.1. Measures by Domain

All screening tools and measures are listed by domain, and where appropriate, by subdomain, in Table 1. Where appropriate, articles describing validation are cited, and examples of which older populations the tools were validated in, as well as examples of comparator instruments used to assess concurrent validity, are presented.

#### 3.1.1. Socio-Demographics

Socio-demographics are assessed using the PANINI Questionnaire and include age, ethnicity, nationality, language, education, and occupation as well as social aspects, including living situation and social circumstances.

#### 3.1.2. General Health

Intoxicants are self-reported and include alcohol, smoking and drug use (quantity and frequency) assessed by the PANINI Questionnaire. Medical history and medication use are also assessed by the PANINI Questionnaire and include self-reported past and current medical conditions and self-reported current medication use, respectively (medical records may be used to supplement self-reported information). Anthropometrics are objectively assessed using a calibrated height and weight measuring system, from which measures of height (cm) and weight (kg) can be obtained and body mass index (BMI) can be calculated (weight (kg)/[height (m)]^2^). Dual-energy X-ray absorptiometry (DXA) [13,14,15] and Bioelectrical Impedance Analysis (BIA) [13,16,17] are included as tools for objective assessment of body composition from which measures of fat mass, lean soft-tissue mass (comprising muscle, inner organs and body water), and bone mineral content can be obtained (e.g., appendicular lean mass (ALM) and skeletal muscle index (SMI)).

#### 3.1.3. Nutrition

Malnutrition is assessed by the Mini Nutritional Assessment (MNA) questionnaire, which is measured in points and stratifies risk of malnutrition as normal nutrition status, at risk of malnutrition and malnourished [18]. Dietary intake (quantity and frequency) is assessed by a food frequency questionnaire (FFQ) [19] to ascertain measures of macronutrient and micronutrient intake via self-report.

#### 3.1.4. Physical Activity and Physical Performance

Physical activity is assessed by two self-reported assessment tools, the Modified Minnesota Leisure Time Activities (MLTA) Questionnaire [20,21] and the International Physical Activity Questionnaire Short Form (IPAQ-s) [22,23], to ascertain self-reported physical activity in kcals/week and hours/day, respectively. Accelerometry is included for objective assessment of physical activity and to obtain measures of physical activity level, energy expenditure and sedentary behavior. Mobility is captured by the PANINI Questionnaire, which includes questions pertaining to self-reported mobility. The Short Physical Performance Battery (SPPB), balance tests with eyes closed and handheld dynamometers are included as tools for the physical performance. The SPPB measures include gait speed (m/s) obtained from a four-meter walk test, time to complete 5 stands from a chair (s) from the chair stand test and ability to maintain balance for 10 s from side-by-side, tandem, and semi-tandem balance tests, as well as a composite score from these three individual tests [24]. Handgrip strength (kg) is assessed using a handheld dynamometer [25,26,27,28]. Frailty is assessed using the Fried Frailty Phenotype, which includes five tools for five criteria to ascertain frailty status: self-reported unintentional weight loss to assess shrinking; hand grip (kg) assessed using a handheld dynamometer to identify weakness; 4 m walk test to assess slowness using a cut-off for gait speed (m/s); Minnesota Leisure Time Physical Activity (MLTPA) questionnaire to assess low activity determined by kcal/week cut-off; and two questions from the Center for Epidemiological Studies depression (CES-D) scale to assess self-reported exhaustion indicating poor endurance and energy [29]. Falls are assessed using the Short Fall Efficacy Scale International (FES-I) to evaluate fear of falling and using the PANINI Questionnaire to assess fall history information in the past year [30]. Activities of daily living (ADLs) are assessed by the Katz Index of Independence in activities of daily living (ADL) and measures the ability to independently complete ADLs via self-report [31,32,33].

#### 3.1.5. Psychological and Cognitive Health

Cognition is assessed through the Standardized Mini Mental State Examination (SMMSE), which is an interviewer-administered screening test for cognitive impairment covering a range of cognitive domains [34,35]. Depression is assessed by the Geriatric Depression Scale-15 (GDS-15), which screens for depression by identifying absence of depression, mild depression, moderate depression and severe depression [36,37,38].

#### 3.1.6. The PANINI Toolkit Protocol and PANINI Case Report Form (CRF)

All tools and measures in the toolkit include standardized measurement schemes, definitions and scoring and are described in further detail in the PANINI Toolkit Protocol (Appendix A) and PANINI CRF (Appendix A).

### 3.2. Application of the PANINI Toolkit

The PANINI Toolkit was adapted and refined over several months and launched in October 2016 for implementation to aid in uniformity of new data collected across consortium-wide research projects. The PANINI Toolkit has been applied across data collection projects within the PANINI consortium (as appropriate based on study design) and aims to ultimately allow for comparability between different ageing populations and various pre- and post-physical activities and nutritional interventions by using the same protocol of measurements [39,40]. Data management for all projects adheres to the FAIR principles [41].

## 4. Discussion

The PANINI Toolkit is a set of tools and measures that provides a comprehensive and multidimensional assessment of health of older adults focusing on the domains of PA and nutrition. The toolkit brings together five domains representing different facets of health in older adults that contribute to overall health status and can interact with one another. Subsequently, it provides a standardized and comprehensive approach towards assessment of health in older adults for research purposes that supports interoperability and re-use of data.

It was a prerequisite for the PANINI Toolkit to be applicable in existing and new datasets of diverse populations, which include (but are not limited to) frail inpatients, care home residents, older adults with a recent hip fracture, patients undergoing elective hip/knee surgery, acute hospital in-patients with a range of morbidities, menopausal women, older adults from ethnic minority groups and healthy community-dwelling older adults. Further, deliberation was conducted by experts across different biomedical fields and nationalities, who each had their own experiences and preferences in using various tools, which is reflected in the decision-making process and shaped its overall result as a multidisciplinary and cross-culturally representative toolkit. It is important to note that the toolkit is intended to provide a recommended framework for research and aid in data collection. Researchers apart of the PANINI consortium were encouraged to use the PANINI Toolkit as it fit with their study design, aims and objectives [39,40].

The PANINI Toolkit consists of validated measures with an emphasis on applicability in research and clinical practice. Despite the deliberate selection of measures in the present toolkit, it is important to acknowledge that there are a breadth of tools for assessing each of these domains [13,18,42,43,44,45]. The tools in the PANINI Toolkit were chosen with the requirement of having a wide bandwidth for use in different populations and settings. However, the use of tools that are intended to be applied generally can lead to floor effects (e.g., frail populations) and ceiling effects (e.g., very healthy populations) [46,47]. Self-reported measures were included in light of their practical benefits, but it is important to acknowledge that these assess perception rather than actual status [48,49].

The incorporation of a wide range of domains and clinically relevant tools aligns with that of other methods that support the concept that the health of older adults is multidimensional, such as the International Classification of Functioning, Disability and Health (ICF) [50] and comprehensive geriatric assessments (CGAs) [43,51]. The PANINI Toolkit focuses on modifiable lifestyle factors with nutrition and physical activity as the core domains, and it is intended to be used primarily for research purposes. This emphasis on nutrition and physical activity is appropriate for the purposes of the PANINI Toolkit and the objective of combining data from new and existing datasets across the PANINI consortium. The current focus of the PANINI Toolkit does not rule out any extension of included domains or incorporation of other domains, e.g., social and environmental domains as covered by the ICF.

The use of standardized tools is particularly important in the assessment of health outcomes within and between the included domains because it facilitates coordination and communication between researchers and clinicians from different fields. In addition, it allows for increased generalizability and aids in the synthesis of findings across different populations, interventions and research projects. The previous literature in the geriatric field has often cited the lack of shared methodology as a limitation and barrier to research [52,53,54]. It has been shown that the use of different screening tools for one clinical outcome (e.g., sarcopenia, malnutrition, frailty), in the same population can arrive at different estimates of prevalence [55,56]. Subsequently, this toolkit provides a method of fostering standardized datasets with shared definitions, methodology, scoring systems and outcome measures that can be used in coordination to address the challenges of ageing.

Although this toolkit contains exclusively validated measures (with the exception of the PANINI Questionnaire), the toolkit as a whole has not yet been validated for use. Application in ongoing PANINI projects across Europe has shown the feasibility of its use within research endeavors; however, the feasibility of the PANINI Toolkit within a non-research or clinical framework has not yet been tested. Further research may be required to validate the toolkit as a whole, evaluate cost-effectiveness and demonstrate wide execution. To our knowledge, this is the first nutrition and physical activity toolkit for research purposes that has been applied simultaneously across European projects. Further research needs to be carried out in each of these domains individually, as well as for their interactions, and it would be beneficial to the research field if these studies were conducted using a standardized comprehensive approach as described in this article.

## Figures and Tables

**Table 1 healthcare-10-01017-t001:** Summary of the PANINI Toolkit tools and measures by domain.

Domain	Tool	Measure(s)	Examples of Validation in Older Adults		
Subdomain			Article	Population(s)	Comparator
**(1) Socio-demographics**					
Socio-demographic	PANINI Questionnaire	Age, nationality, language, education and occupation	n/a	n/a	n/a
Social	PANINI Questionnaire	Marital status, living situation and social circumstances	n/a	n/a	n/a
**(2) General Health**					
Intoxicants	PANINI Questionnaire	Alcohol, smoking and drug use	n/a	n/a	n/a
Medical history	PANINI Questionnaire	Past and current medical conditions	n/a	n/a	n/a
Medication use	PANINI Questionnaire	Current medication use	n/a	n/a	n/a
Anthropometrics	Calibrated height and weight measure	Height (cm) and weight (kg)	n/a	n/a	n/a
	Tape measure	Waist, hip, calf and mid-arm circumferences (cm)	n/a	n/a	n/a
Body composition	Dual-energy X-ray absorptiometry (DXA)	Fat mass, lean soft-tissue mass (comprising muscle, inner organs and body water) and bone mineral content (kg) (Same measures as DXA)	[13,16,57]	CD, H	4-C-model, CT, MRI
	Bioelectrical Impedance Analysis (BIA)		[17,58,59]	CD, H	TBW, 4-C model, CT, DXA
**(3) Nutrition**					
Malnutrition	Mini Nutritional Assessment (MNA)	Nutritional status (points) (normal nutritional status, at risk of malnutrition, malnourished)	[18,60,61]	CD, H, I, Frail, Healthy	Nutritional assessment by physician, comprehensive nutritional assessment (anthropometry, biochemistry and dietary intake) other malnutrition screening tools
Dietary intake	Food frequency questionnaire	Macronutrient and micronutrient intake (incl. fluid intake)	[19]	CD	24 h recalls
**(4) Physical Activity and Physical Performance**					
Physical activity	Modified Minnesota Leisure Time Activities (MLTPA) Questionnaire	Self-reported physical activity (kcal/week)	[20,21]	CD	Accelerometer
	International Physical Activity Questionnaire Short Form (IPAQ-s)	Self-reported physical activity (vigorous, moderate, walking, sitting) (hours/day)	[22,23]	CD	Accelerometer
	Accelerometer	Physical activity, energy expenditure and sedentary behavior	n/a	n/a	n/a
Mobility	PANINI Questionnaire	Self-reported mobility	n/a	n/a	n/a
Physical Performance	Short Physical Performance Battery (SPPB)	Composite physical performance score (points)	[24,62,63,64,65]	CD, H	Self-reported mobility limitations/disability
	4 m walk test	Gait speed (m/s)			
	Chair stand test	Time to complete 5 stands from a chair (s)			
	Balance tests	Ability to maintain balance for 10 s in side-by-side, tandem and semi-tandem positions (yes/no, points)			
	Balance tests with eyes closed	(Same as SPPB balance tests with eyes closed)			
	Handheld dynamometer	Hand grip strength (kg)	[25,26,28,65]	CD	Isometric muscle strength (knee extension, hip flexor, elbow flexion, trunk extension, pinch)
Frailty	Fried Frailty Phenotype Criteria	Frailty status (robust, pre-frail or frail) ^a^	[29,66,67]	CD	Other frailty indices (e.g., Frailty Index)
	Shrinking: weight loss questions ^b^	Shrinking: unintentional weight loss in past year ^b^ (yes/no)			
	Weakness: handheld dynamometer	Weakness: hand grip strength (kg)			
	Poor endurance: Depression Center for Epidemiologic Studies Depression Scale (CES-D) ^c^	Poor endurance: depression score (points)			
	Slowness: 4 m walk test	Slowness: gait speed (m/s)			
	Low activity: MLTPA Questionnaire	Low Activity: physical activity (kcal/week)			
Falls	Short Fall Efficacy Scale International (FES-I)	Fear of falling (points)	[30]	CD	FES-I (original)
	PANINI Questionnaire	Fall history information in past year			
Activities of daily living (ADLs)	Katz Index of Independence in Activities of daily living (ADL)	Ability to independently complete ADLs (yes/no, points)	[31,32,33]	CD	Self-reported mobility impairment/disability
**(5) Psychological and Cognitive Health**					
Cognition	Standardized Mini Mental State Examination (SMMSE)	Cognitive status (points)	[34,35,44]	I, H	MMSE
Psychological	Geriatric Depression Scale-15 (GDS-15)	Depression (normal, mild/moderate/severe depression)	[36,37,38]	CD, H	Structured clinical interviews for DSM-IV and ICD-10 criteria

Bulleted items represent individual measures within a composite measure. Measures from the PANINI Questionnaire are assessed through customized questionnaire. m = meter, 4-c = 4-component, CT = computed tomography, MRI = magnetic resonance imaging, TBW = total body water, CD = community dwelling, H = hospitalized, I = institutionalized, incl. = including, n/a = not applicable. ^a^ All criteria of the Fried Frailty Phenotype are assessed as above or below a specified threshold cutoff (adjusted for BMI, age, height and/or sex as appropriate). ^b^ Unintentional weight loss is assessed by self-reported weight loss of ≥4.5kg in the year before the current evaluation, or unintentional weight loss of ≥5% of the previous year’s body weight is used to assess the “shrinking” criteria of the Fried Frailty Phenotype. ^c^ The evaluation of two statements from the CES-D scale: (a) “I felt that everything I did was an effort” and (b) “I could not get going”, is used to assess the “poor endurance” criteria of the Fried Frailty Phenotype.

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
