# Peer review of "The Physical Activity and Nutritional INfluences in Ageing (PANINI) Toolkit: A Standardized Approach towards Physical Activity and Nutritional Assessment of Older Adults"

_healthcare, 2022, doi:10.3390/healthcare10061017_

Round 1
Reviewer 1 Report
The toolkit is a listing of measures to assess multiple dimensions of health with a focus on physical activity and nutrition in older adults. It was designed by expert consensus with an aim to facilitate collection, reuse, and exchangeability of data to enable coordination of research, generalizability of findings, and improve healthcare delivery to older adults. It has not been validated or widely executed. In my opinion, the manuscript could be improved by stating at the outset that this is a report of toolkit development and that additional research will be undertaken to evaluate its reliability and actual value in use. This is done to some extent, but I believe a more forthright would be useful.
Author Response
Manuscript ID: Healthcare-1648950
The Physical Activity and Nutritional INfluences In Ageing (PANINI) Toolkit: A Standardized Approach Towards Physical Activity and Nutritional Assessment of Older Adults
We thank the editor and reviewers for their valuable and constructive comments. We have modified the manuscript according to the reviewers’ suggestions, which we think improved the manuscript considerably. Modifications are listed below. Reviewer comments are written in ‘bold font’. Our response is given in ‘normal font’ and the changes made to revise the text are given in ‘italic font’. Modifications in the manuscript have been added using track-changes.
REVIEWER #1
The toolkit is a listing of measures to assess multiple dimensions of health with a focus on physical activity and nutrition in older adults. It was designed by expert consensus with an aim to facilitate collection, reuse, and exchangeability of data to enable coordination of research, generalizability of findings, and improve healthcare delivery to older adults. It has not been validated or widely executed. In my opinion, the manuscript could be improved by stating at the outset that this is a report of toolkit development and that additional research will be undertaken to evaluate its reliability and actual value in use. This is done to some extent, but I believe a more forthright would be useful.
Thank you for your compliments and critical suggestions regarding our manuscript. We agree that stating early in the manuscript focuses on the toolkits development and content at the outset of the manuscript would aid in transparency. We also agree that reiterating that the manuscript has not been validated nor widely executed is important for clarity. We have adapted the manuscript accordingly.
Changes made (Introduction, page 1 paragraph 3): This article describes the development of The PANINI Toolkit including its content and application within the PANINI consortium.
Changes made (Introduction, page 6, paragraph 6): Further research may be required to validate the toolkit as a whole, evaluate cost-effectiveness and demonstrate wide execution.

Reviewer 2 Report
The aim of this short paper is to present a toolkit that would aim to standardise research initiatives in different populations within the PANINI consortium.
Much of the information described in this informative article has already been presented in other published papers such as: https://dspace.stir.ac.uk/bitstream/1893/29892/1/AGMR_1043.pdf
There is little innovative information in this submitted paper that would be of interest to the scientific community. The toolkit offers no scientific insights of interest or usefulness in clinical practice. It is a set of validated surveys summarised in a toolkit. The important detail is that it is the doctor who decides whether one test or another, one procedure or another, mey be useful. This toolkit is likely to increase medical costs because it incorporates a number of investigations that could be avoided; a doctor experienced in nutritional assessment is perfectly capable of understanding whether it is necessary to prescribe a DEXA or do an MNA in a patient. The toolkit generalises a clinical practice by reducing the doctor's discretionary ability.
Author Response
Manuscript ID: Healthcare-1648950
The Physical Activity and Nutritional INfluences In Ageing (PANINI) Toolkit: A Standardized Approach Towards Physical Activity and Nutritional Assessment of Older Adults
We thank the editor and reviewers for their valuable and constructive comments. We have modified the manuscript according to the reviewers’ suggestions, which we think improved the manuscript considerably. Modifications are listed below. Reviewer comments are written in ‘bold font’. Our response is given in ‘normal font’ and the changes made to revise the text are given in ‘italic font’. Modifications in the manuscript have been added using track-changes.
REVIEWER #2
The aim of this short paper is to present a toolkit that would aim to standardise research initiatives in different populations within the PANINI consortium. Much of the information described in this informative article has already been presented in other published papers such as: https://dspace.stir.ac.uk/bitstream/1893/29892/1/AGMR_1043.pdf
There is little innovative information in this submitted paper that would be of interest to the scientific community. The toolkit offers no scientific insights of interest or usefulness in clinical practice. It is a set of validated surveys summarised in a toolkit. The important detail is that it is the doctor who decides whether one test or another, one procedure or another, mey be useful. This toolkit is likely to increase medical costs because it incorporates a number of investigations that could be avoided; a doctor experienced in nutritional assessment is perfectly capable of understanding whether it is necessary to prescribe a DEXA or do an MNA in a patient. The toolkit generalises a clinical practice by reducing the doctor's discretionary ability.
Thank you for your critical comments and we hope that this revision has improved our manuscript and enhanced its contribution to the scientific community. We have added to the discussion that the toolkit has not been proven cost effective to provide transparency regarding this important consideration you have brought up. Further, it is important to note that the toolkit is meant to be used strategically as a tool and allows for discretion to utilize measures based on needs and appropriateness. Meaning all measures within the toolkit do not need to be used and cost-effectiveness could be taken into consideration if that was a particular concern. However, there is evidence that in geriatric comprehensive assessment rather than fragmented assessment of health conditions has been beneficial and effective without increasing costs of care (doi:10.1016/j.critrevonc.2003.06.005). The toolkit in itself is not intended to impede or reduce a clinician's discretion but to provide an aid for assessment of health status in older adults that is standardized amongst researchers and clinicians so they do not have to search themselves for tools.
Changes made (Discussion, page 6, paragraph 6): Further research may be required to validate the toolkit as a whole, evaluate cost-effectiveness and demonstrate wide execution.

Reviewer 3 Report
This is a highly relevant project developing a standardized toolkit to assess physical activity and nutrition in older adults with the aim to increase data integration and generalizability of findings. It would be helpful to read more about the procedure how measures were chosen, including how consensus was reached and why certain measures were preferred over others. The authors state that the toolkit has been applied across data collection projects within the PANINI consortium, please show results or references here. There are inconsistencies in the discussion, it is emphasized that it is intended to be primarily used in research, but shall facilitate coordination and communication in researchers and clinicians from different fields. Why is intended to be primarily used in research and not in clinical routine? In the manuscript and CRF it is stated that the IPAQ-s assesses physical activity in hours/day, in the protocol it is suggested to use MET-minutes/day or preferably MET-minutes/week. What was the reason to include only ADL and not IADL? In the patient history form, what was the rationale to choose these diseases? For the measurement of circumferences, why do subjects have to put on anti-slip socks (mentioned in the CRF but not protocol)? For measurement of the handgrip strength, do the arms really have to be parallel to the body (CRF)? When norms are given (e.g., for the 4 meter walk), please state the reference for the norms. For the balance test with eyes closed, no scores are given, why? Regarding the dietary intake on page 14 of the CRF, how to you calculate the amounts of macronutrients, minerals and vitamins? Regarding marital status, why is having a partner without being married not included? In the protocol, what does “The protocol for dietary intake will be available after the food frequency questionnaire is confirmed among the beneficiaries” imply? Clarify, how low physical activity as frailty criterion is defined.
Minor:
Typos on page 4 of the manuscript in the socio-demographics section.
Typos on page 5 of the manuscript line 3 in the discussion section.
Mind a consistent naming, “The PANINI Toolkit” vs “the PANINI Toolkit”
Heading of Table 1 in the manuscript is not correct
Typos at the end of page 12 in the CRF
In the protocol in the description of the Short Physical Performance Battery a reference source is missing
Typo in the protocol in the description of the additional question for water consumption
Author Response
Manuscript ID: Healthcare-1648950
The Physical Activity and Nutritional INfluences In Ageing (PANINI) Toolkit: A Standardized Approach Towards Physical Activity and Nutritional Assessment of Older Adults
We thank the editor and reviewers for their valuable and constructive comments. We have modified the manuscript according to the reviewers’ suggestions, which we think improved the manuscript considerably. Modifications are listed below. Reviewer comments are written in ‘bold font’. Our response is given in ‘normal font’ and the changes made to revise the text are given in ‘italic font’. Modifications in the manuscript have been added using track-changes.
REVIEWER #3
This is a highly relevant project developing a standardized toolkit to assess physical activity and nutrition in older adults with the aim to increase data integration and generalizability of findings. It would be helpful to read more about the procedure how measures were chosen, including how consensus was reached and why certain measures were preferred over others. The authors state that the toolkit has been applied across data collection projects within the PANINI consortium, please show results or references here. There are inconsistencies in the discussion, it is emphasized that it is intended to be primarily used in research, but shall facilitate coordination and communication in researchers and clinicians from different fields. Why is intended to be primarily used in research and not in clinical routine? In the manuscript and CRF it is stated that the IPAQ-s assesses physical activity in hours/day, in the protocol it is suggested to use MET-minutes/day or preferably MET-
minutes/week. What was the reason to include only ADL and not IADL? In the patient history form, what was the rationale to choose these diseases? For the measurement of circumferences, why do subjects have to put on anti-slip socks (mentioned in the CRF but not protocol)? For measurement of the handgrip strength, do the arms really have to be parallel to the body (CRF)? When norms are given (e.g., for the 4 meter walk), please state the reference for the norms. For the balance test with eyes closed, no scores are given, why? Regarding the dietary intake on page 14 of the CRF, how to you calculate the amounts of macronutrients, minerals and vitamins? Regarding marital status, why is having a partner without being married not included? In the protocol, what does “The protocol for dietary intake will be available after the food frequency questionnaire is confirmed among the beneficiaries” imply? Clarify, how low physical activity as frailty criterion is defined.
Minor:
Typos on page 4 of the manuscript in the socio-demographics section.
Typos on page 5 of the manuscript line 3 in the discussion section.
Mind a consistent naming, “The PANINI Toolkit” vs “the PANINI Toolkit” Heading of Table 1 in the manuscript is not correct Typos at the end of page 12 in the CRF
In the protocol in the description of the Short Physical Performance Battery a reference source is missing
Typo in the protocol in the description of the additional question for water consumption
We appreciate your appraisal of our research and critical comments. We think the manuscript can be substantially improved from the suggestions you have made. We agree that the selection of the measures can be further clarified and have adapted the manuscript to do so.
Changes made (Methods, page 3, paragraph 2): Consensus was reached through discussion and any disagreements were resolved through evaluation of the aforementioned consideration factors and similar tools/measures were included when the added value of including both could be justified and agreed upon.
We agree that it is important to demonstrate application of the PANINI Toolkit and have adapted the manuscript to provide further information on where the PANINI Toolkit is being used. As not all papers using the PANINI Toolkit have been published, we have also added the references of the PANINI consortium wide papers, that provide a summary of all research activities within the consortium and make use of the PANINI Toolkit.
Whittaker, A. C., Delledonne, M., Finni, T., Garagnani, P., Greig, C., Kallen, V., et al. (2018). Physical Activity and Nutrition INfluences In ageing (PANINI): consortium mission statement. Aging clinical and experimental research, 30(6), 685–692. https://doi.org/10.1007/s40520-017-0823-7
Whittaker, A.C., Asamane, E.A., Aunger, J.A., Bondarev, D., Cabbia, A., Doody, P.D., et al. (2019). Physical Activity and Nutrition INfluences in Ageing: Current Findings from the PANINI Project. Advances in Geriatric Medicine and Research, 1. e190005. https://doi.org/10.20900/agmr20190005
Changes made: References added throughout.
The PANINI Toolkit was intended for research purposes rather than clinical practice based on the goals of the consortium in coordinating multidisciplinary research endeavors across the EU. The highlighting of the toolkit’s potential benefit to clinicians was done to reflect deliberate choices in measure selection to enhance its utility beyond research. That is to say that the toolkit was designed to incorporate only measures that were clinically relevant to help bridge the gap between research and clinical practice and further to aid in interpretation of findings (familiarity with measures used) and potential use for clinicians so they would not have to search themselves for tools. We do not preclude that fact that the PANINI toolkit may be even more useful for use in clinical practice, however, we are cautious to make claims regarding its utility in this setting as it has not been tested and issues such as cost-effectiveness may be more important to consider before advocating its use. We have clarified this.
We appreciate comments regarding the content of the PANINI protocol and CRF; however, as the PANINI Toolkit has already been launched and used, we can unfortunately no longer make changes to the content. The toolkit is intended to serve as a tool and things such as the units, protocol, questionnaire items, can be adapted as necessary based on the needs of the user but to optimize standardization we provide a protocol and CRF to be used as described.
Thank you for your minor comments and detailed assessment - we have made all corrections suggested.
Changes made: Throughout manuscript

Round 2
Reviewer 2 Report
The authors made only minor changes to the paper. The serious shortcomings reported in the first review remain.
This is an informative paper with no science. The number of authors is shamefully high for such an uninnovative paper.
Much of the information described in this informative article has already been presented in other published papers such as: https://dspace.stir.ac.uk/bitstream/1893/29892/1/AGMR_1043.pdf
There is little innovative information in this submitted paper that would be of interest to the scientific community. The toolkit offers no scientific insights of interest or usefulness in clinical practice. It is a set of validated surveys summarised in a toolkit.
Author Response
We agree with the reviewer that the tools being used are not new. During the last decades multiple tools have been developed and no standardization is in place regarding which tools should be used to phenotype older individuals. Within the PANINI consortium consisting of international experts in the filed, we extensively discussed which tools to use, as described in the manuscript. This should form the basis for further guidance of researchers and clinicians working with older individuals.
Reviewer 3 Report
The authors adequately responded to all points raised by the reviewer.
Author Response
We appreciate the reviewer's second review and positive comments regarding our revision and responses.